# N-Acetylcysteine Antagonizes NGF Activation of TrkA through Disulfide Bridge Interaction, an Effect Which May Contribute to Its Analgesic Activity

**DOI:** 10.3390/ijms25010206

**Published:** 2023-12-22

**Authors:** Stefano Govoni, Piercarlo Fantucci, Nicoletta Marchesi, Jacopo Vertemara, Alessia Pascale, Massimo Allegri, Laura Calvillo, Emilio Vanoli

**Affiliations:** 1Department of Drug Sciences, Pharmacology Section, University of Pavia, 27100 Pavia, Italy; govonis@unipv.it (S.G.); nicoletta.marchesi@unipv.it (N.M.); 2Department of Biotechnology and Biosciences, University of Milan-Bicocca, 20126 Milan, Italy; piercarlo.fantucci@unimib.it (P.F.); jacopo.vertemara@unimib.it (J.V.); 3Centre Lémanique de Neuromodulation et Thérapie de la Douleur, Hôpital de Morges, Ensemble Hospitalier de la Côte (EHC), 1110 Morges, Switzerland; massimo.allegri@ehc.vd.ch; 4Department of Cardiology, Cardiology Research Laboratory, Istituto Auxologico Italiano IRCCS, 28824 Milan, Italy; l.calvillo@auxologico.it; 5School of Nursing, University of Pavia, 27100 Pavia, Italy; emilio.vanoli@unipv.it

**Keywords:** TrkA, pain, N-acetylcysteine, NGF

## Abstract

N-acetylcysteine (NAC), a mucolytic agent and an antidote to acetaminophen intoxication, has been studied in experimental conditions and trials exploring its analgesic activity based on its antioxidant and anti-inflammatory properties. The purpose of this study is to investigate additional mechanisms, namely, the inhibition of nerve growth factor (NGF) and the activation of the Tropomyosin receptor kinase A (TrkA) receptor, which is responsible for nociception. In silico studies were conducted to evaluate dithiothreitol and NAC’s interaction with TrkA. We also measured the autophosphorylation of TrkA in SH-SY5Y cells via ELISA to assess NAC’s in vitro activity against NGF-induced TrkA activation. The in silico and in vitro tests show that NAC interferes with NGF-induced TrkA activation. In particular, NAC breaks the disulfide-bound Cys 300–345 of TrkA, perturbing the NGF-TrkA interaction and producing a rearrangement of the binding site, inducing a consequent loss of their molecular recognition and spatial reorganization, which are necessary for the induction of the autophosphorylation process. The latter was inhibited by 40% using 20 mM NAC. These findings suggest that NAC could have a role as a TrkA antagonist, an action that may contribute to the activity and use of NAC in various pain states (acute, chronic, nociplastic) sustained by NGF hyperactivity and/or accompanied by spinal cord sensitization.

## 1. Introduction

Several lines of evidence point to nerve growth factor (NGF) and its signaling pathway through the Tropomyosin receptor kinase A (TrkA) receptor as an important player in mechanisms sustaining various types of pain, including osteoarthritic, low back, and neuropathic pain [1,2,3,4]. Indeed, genetic studies in animals and humans suffering from congenital pain insensitivity and peripheral sensory neuropathy underscore the involvement of the NGF pathway [5,6]. The action of NGF is complex and may impact acutely both in sensitizing to pain, through actions involving glutamate and Transient Receptor Potential Vanilloid 1 (TRPV1) receptors [7], and with more indirect immune-mediated mechanisms involving the NGF-induced production of prostaglandin D2 (PGD2) in joint mast cells [8], both of them increasing the risk of developing chronic pain. Accordingly, several anti-NGF strategies have been developed among which are the use of monoclonal antibodies against NGF and of small antagonist molecules at the TrkA receptor [9,10,11,12]. In the latter case, the search for peptides capable of blocking the interaction of NGF with TrkA is of particular interest as a possible antinociceptive strategy [12]. It should be noted that the cysteine residue at the C-terminus of such peptides is required to inhibit the NGF/TrkA interaction, despite not being involved in NGF binding, and that peptide dimerization enhances the inhibitory effect [12]. Interestingly enough, in previous research on novel peptide compounds with TrkA agonist/antagonist activities, Fantucci et al. [13], searching for the smallest representative of the TrkA-NGF complex, found that the highest number of short-range interactions occur between NGF and TrkA, involving, respectively, the amino acid residues from 2 to 24 of NGF and the sequence 250–348 of TrkA, where the latter sequence contains a disulfide bond between cysteine 300 and cysteine 345. While assaying the activity of synthesized selected peptides from the series proposed by Fantucci et al. [13], in a cell-based TrkA autophosphorylation assay, we realized that reducing agents such as dithiothreitol (DTT) produced a dose-dependent inhibition of the TrkA autophosphorylation induced by NGF, an interaction that was also confirmed in silico (see below in the Results). These observations prompted us to test other compounds with the ability to disrupt disulfide bonds. In particular, we decided to test in silico and in vitro N-acetylcysteine (NAC) in order to evaluate its ability to interfere with the NGF activation of TrkA. Indeed, NAC is a well-known mucolytic agent with a long history of use and safety in the pulmonary diseases field with expanding applications in various medical conditions [14]. NAC has well-known antioxidant/reducing properties and is capable of interacting with disulfide bonds [15], is widely available, and is commonly used for antianalgesic purposes [16].

## 2. Results

### 2.1. In Silico Studies

The basic molecular interaction considered is that occurring between the thiol group of the docking species (HSd; as DTT or NAC) with the Sp-Sp atomic group (Sp stands for a sulfur atom belonging to the protein), where SdH····Sp-Sp (Mode A) corresponds to the formation of a H-bond and a consequent weakening of the Sp-Sp bond. Eventually, the SdH····Sp-Sp may be strong enough to promote the breaking of the Sp-Sp bond according to SdH····Sp-Sp → Sd-Sp····HSp (Mode B). This is carried out using the CovDock tool of MAESTRO [17]. DTT is known to be quite a strong reducing agent characterized by two -SH terminal groups, both available as reducing sites, with the formation of a cycle characterized by an internal Sd-Sd bond. Such a tendency is more pronounced when the dimeric or trimeric forms of DTT are considered, which can give a more stable cycle with a larger number of atoms. The docking results show that growing the DTT chain is associated with an increased affinity and that all the polymers bind to the same site. In particular, Figure 1A shows the crystallographic complex of TrkA (red and gray structures) and NGF (yellow and orange). The atomic footprint of mono-DTT, bi-DTT, and tri-DTT molecules is reported, respectively, in red, yellow, and blue (left portion of Figure 1A, the same color code is also used in the right square of Figure 1A). The binding of DTT is near the two cysteines forming the disulfide bond (right square of Figure 1A). This observation, together with the strong reducing activity of DTT, allows one to speculate that the observed inhibition of TrkA activation by NGF may be due to the unfolding of TrkA caused by the breaking of the disulfide bond. The docking results of TrkA-DTT (Mode A), reported in panel B of Figure 1, show that the binding energy increases in the order mono-DTT < bis-DTT < tris-DTT. However, the absolute values of the computed interactions are relatively small (3.0–5.0 kcal/mole) when compared with the corresponding results for TrkA-NAC interaction, which are equal to 12.8 kcal/mole. The docking results were further refined through the calculation of binding free energies (Δ*G_binding_*) using the MM-GBSA method, validating that Tris-DTT has the highest affinity for TrkA among the tested molecules. Due to the evidence of a stronger interaction computed for NAC-TrkA, we also considered the Mode B docking simulation. The results are reported in Figure 2A,B in which, for completeness, both the Mode A and Mode B poses are shown. Figure 2A is well representative of a H-bond involving both of the Sp-Sp atoms, while Figure 2B clearly shows the formation of the Sd-Sp bond between the NAC and Cys345 and the protonation of the sulfur atom of Cys300. Therefore, the results of this simulation show that covalent bonds are involved in breaking and formation, respectively, with a global free energy of reaction (which is different from the binding energy discussed above) equal to 31.1 kcal/mole, which is much larger than the value computed when only van der Waals and Coulomb interactions are considered. This also shows that the TrkA-NAC interaction is really able to break the Sp-Sp, thus producing a structural rearrangement internal to the binding site and a consequent loss of the molecular recognition capability of TrkA-NGF.

### 2.2. In Vitro Studies in Cultured Neuroblastoma Cells

The exposure of SH-SY5Y neuroblastoma cells for 10 min. to 100 ng/mL NGF produced an increased TrkA autophosphorylation that was inhibited by 60% using 0.4 mM DTT and almost fully abolished by 10 mM DTT, a concentration that displayed some toxicity as assessed by the measure of the cell mitochondrial activity (Figure 3A,B).

We then studied the effect of NAC on TrkA autophosphorylation elicited by NGF. Figure 4 shows the ability of various concentrations of NAC, from 1.0 to 50 mM, to inhibit TrkA activation by NGF (100 ng/mL). The effect of NAC (Figure 4A) was concentration-dependent, reaching a 60% inhibition at 50 mM, a concentration that was toxic for the cells, as shown in Figure 4B by the results of the MTT assay. At the concentration of 20 mM, not affecting viability, NAC inhibited TrkA activation by 40% (*p* < 0.0001). This inhibition was similar to the one observed with micromolar concentrations of the two published small-molecule TrkA inhibitors (AG879 and RO08-2750) used when setting up the method (see Appendix A).

## 3. Discussion

The results show that in silico NAC was able to bind to the TrkA receptor with a strength of binding equal to 12.8 kcal/mole. In addition, in vitro NAC interfered with the activation of TrkA by NGF in SH-SY5Y neuroblastoma cells at millimolar concentrations not affecting cell viability (up to 20 mM).

Using 20 mM NAC, the inhibition extent was 40% and statistically significant (Figure 4). Based on these results and the fact that NAC has more reducing power compared to DTT, the inhibitory effect of the former molecule is due to the reduction of the disulfide bridge, as observed with the reducing molecule DTT (Figure 1) and also as confirmed by the in silico data of NAC (Figure 2).

This novel activity of NAC, which inhibits TrkA activation by NGF, possibly contributes, along with its antioxidant properties, to its analgesic effect in various experimental settings and models, as summarized in Table 1.

Indeed, NAC’s analgesic action may be due to the antioxidant action of the compound. This is due to its ability to increase the antioxidant defenses of the organism by sustaining the glutathione pathway. Also, NAC’s ability to inhibit the cytokines that participate in the inflammatory response [22] may explain its effect on pain and, at the pulmonary level, its ability to reduce inflammation [15]. Moreover, since this anti-inflammatory activity occurs without the impairment of the immune response, it could help to reduce acute pain without increasing the risk of chronic pain, unlike nonsteroidal anti-inflammatory drugs (NSAIDs) or steroids, as Parisien et al. [23] have demonstrated recently.

On the other hand, the analgesic activity of NAC has been shown to be effective in conditions that involve less inflammatory processes, as shown in a recent clinical study [19]. Patients undergoing spinal surgery were able to reduce their opioid use by receiving high doses of intravenous NAC intraoperatively. There may also be other mechanisms contributing to its analgesic effects based on the observations of endometriosis. These effects may indeed rely, at least in part, on the action opposing to NGF activation that mediates sensitization to pain at the spinal level. In animals, a previous study [24] suggested that NAC administered via the intrathecal route exerts an analgesic effect in a murine model (formalin test, 65% reduction in the licking time). Using NAC intrathecally is a quick practical way to reach about 20 mM concentrations of NAC that can inhibit TrkA activation by NGF, as indicated in experiments on cultured cells. In fact, the literature data indicates that, even at the highest tolerable intravenous doses, NAC does not reach in the cerebrospinal fluid those millimolar concentrations that may be attained by this route of administration and which are required to acutely modulate TrkA activation by NGF.

The final understanding proposed in this manuscript brings an interaction mechanism between NAC and Trk that is not fully clarified. Our finding is based on a thorough in silico study identifying the specific mechanisms of the NAC inhibition of NGF binding with a precise and defined interaction with two cysteine residues of the TrkA sequence. Further experiments should better clarify whether these are the sole cysteine bridges involved in the action of NAC on TrkA and also whether the p75 receptor for NGF may be involved. Of interest is the fact that, in the conditions adopted, the inhibition of the receptor is not complete; the latter effect is considered as having negative consequences as shown when using NGF monoclonal antibodies in osteoarthritis [25]. Accordingly, the partial inhibition by NAC exerted at a nontoxic cellular concentration may be considered advantageous. Despite the need for in vivo experiments to better define this point, the long-term documented clinical use and tolerability of NAC do not suggest that side effects related to NGF antagonistic activity exist, in particular, when NAC is used in osteoarthritis patients (as reviewed in [16]).

## 4. Materials and Methods

### 4.1. In Silico Studies

The in silico study of the interaction between TrkA and NGF has been carried out, mainly on the basis of molecular docking techniques. As is well known, such procedures basically require identifying a simulation box around the putative receptor site of the protein and the structure of the docking molecule that is allowed to freely rototranslate within the box to locate the most energetically favorable location (the docking pose). In our case, the definition of the receptor site was established as follows. The crystallographic structure of TrkA was retrieved from the PDB file with the code 2IFG downloaded from the Protein Data Bank (https://www.rcsb.org/). The protein structure was further refined before docking studies by applying the Protein Preparation Wizard [26] available in MAESTRO in order to reconstruct any missing loops or unresolved residues, assign bond orders, create disulfide bonds, and generate the proper protonation state of residues at pH 7.0. The grid box with a size of 36 × 36 × 36 Å was generated around the TrkA region where the NGF cocrystallized moiety showed the minimum average distances (the supposed maximum interaction). To check the presence of false positives, a second round of simulations was conducted using a smaller grid (20 × 20 × 20 Å) centered on the protein region of the previous poses. The identified receptor site was considerably larger, and this required us to reduce the flexibility within the receptor, except for the orientation of the OH groups of serine, threonine, and tyrosine, which seem to be necessary for an extended H-bond network.

The docking simulations were conducted with Glide [27] using the SP-peptide mode [28] that is designed to handle the much greater flexibility of peptides relative to the usual kinds of ligands. All the simulations were performed using OPLS2005 [29] as a force field. In the present study, the docking simulations were aimed at identifying the capability of the docking molecule to interact with the -S-S- sulfur bridge between the Cys300 and Cys345 residues. Docking studies were performed for DTT and NAC (see Section 2). The binding energy values reported in the following are expressed in kcal/mole and refer to the best identified docking pose. The Prime/MM–GBSA method was used to evaluate the binding free energy (Δ*G_binding_*) of each ligand pose obtained from docking simulation according to the following equation [30]:$${∆G}_{binding}={∆G}_{MM}+{∆G}_{solv}+{∆G}_{SA}$$
where ${∆G}_{MM}$ is the difference in the minimized energies between the DTT-TrkA complex and the sum of the energies of the unliganded TrkA and DTT; ${∆G}_{solv}$ is the difference in the GBSA solvation energy of the DTT–TrkA complex and the sum of the solvation energies for the unliganded TrkA and DDT; and ${∆G}_{SA}$ is the difference in surface area energies for the complex and the sum of the surface area energies for the unliganded TrkA and DTT.

### 4.2. In Vitro Studies

Human neuroblastoma SH-SY5Y cells were obtained from ATCC (Manassas, VA, USA) and cultured in T75 flasks in a humidified incubator at 37 °C with 5% CO_2_. SH-SY5Y cells were grown in Eagle’s minimum essential medium (EMEM) supplemented with 10% fetal bovine serum, 1% penicillin–streptomycin, L-glutamine (2 mM), nonessential amino acids (1 mM), and sodium pyruvate (1 mM). In MTT experiments, the cells were exposed to 0.5, 1, 5, 10, 20, and 50 mM NAC (Sigma-Aldrich, Darmstadt, Germany) for 90 min. For ELISA experiments, the cells were exposed to various concentrations of NAC and DTT (Sigma-Aldrich, Darmstadt, Germany) for 90 min and then to 100 ng/mL NGF (rh beta-NGF, ImmunoTools, Friesoythe, Germany) for 10 min. The concentration and the time chosen for NGF were the best combination as derived from preliminary experiments using concentrations of NGF ranging from 0.5 to 100 ng/mL and times from 5 to 60 min, eliciting TrkA autophosphorylation response ranging from 2.5 to 3.5 fold the basal rate (see Appendix A). The activation was inhibited by micromolar concentrations of AG879 and RO08-2750 (Tocris Biosscience, Bristol, UK), two described inhibitors of TrkA which are commercially available. The entire experiment was performed under a laminar flow hood.

### 4.3. MTT Assay

Mitochondrial enzymatic activity was estimated via MTT [3-(4,5-dimethylthiazol-2-yl)-2,5-diphenyltetrazolium bromide] assay (Sigma-Aldrich, Darmstadt, Germany). A cell suspension of 2 × 10^4^ cells/mL (for SH-SY5Y cell line) was seeded into 96-well plates. Following each treatment, 50 μL of MTT (concentration equal to 2.5 mg/mL) were added to each well. After incubation at 37 °C for 3 h, the purple formazan crystals were formed. The formed crystals were solubilized in dimethylsulfoxide (DMSO; Sigma-Aldrich, Darmstadt, Germany). Specifically, after removing the MTT from the wells, 100 μL of DMSO was added in order to lyse the cellular and mitochondrial membranes and solubilize the formazan crystals. After 10 min, absorbance values were measured at 595 nm using a Synergy HT microplate reader (BioTek Instruments, Santa Clara, CA, USA), and the results are expressed as % with respect to control.

### 4.4. ELISA Assay

The phosphorylation of TrkA receptor (phosphor-TrkA, pTrkA) in SH-SY5Y was estimated with a specific ELISA kit (Cell Signaling Technologies, Danvers, MA, USA), according to the manufacturer’s instructions. This assay employed the quantitative sandwich enzyme immunoassay technique. A monoclonal antibody specific for phosphor-TrkA was already precoated onto a microplate. Standards and samples were pipetted into the wells, and any phosphor-TrkA present was bound by the immobilized antibody. After washing away any unbound substances, an enzyme-linked polyclonal antibody specific for phosphor-TrkA was added to the wells. Following a wash to remove any unbound antibody, a substrate solution was added, and the color was developed in proportion to the amount of phosphor-TrkA bound in the initial step. The color development was stopped, and the intensity of the color was measured (570/450 nm).

## 5. Conclusions

By acting on cysteine residues 300–345 of the TrkA receptor structure and breaking the disulfide bridge, this paper identifies a novel mechanism for NAC to inhibit TrkA activation by NGF. Since the search for NGF antagonistic activity is actively pursued in the field of pain therapy, the characterization of a novel mechanism using repurposed or ex novo-developed substances containing thiolic-reducing groups may represent a new strategy. Moreover, in cases such as NAC, the repurposed compounds may have long histories of use and safety and may be helpful and used to develop novel strategies to approach several acute and chronic pain conditions.

## 6. Patents

All data are protected by the patent number WO2022223590A.

## Figures and Tables

**Figure 1 ijms-25-00206-f001:**
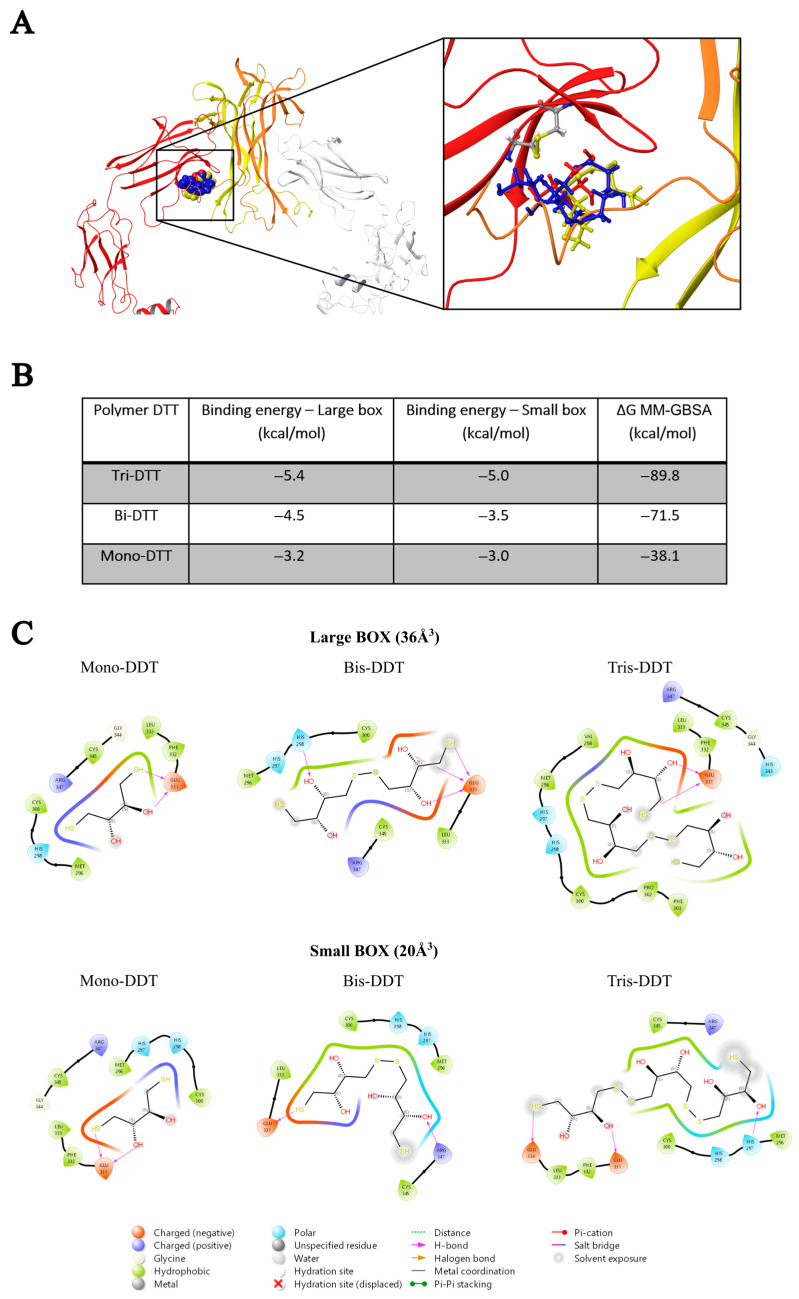
In silico analysis (**A**) of the interaction between DDT and TrkA at the disulfide bond between cysteines 300–345. (**B**) Binding energy of the DTT polymer in kcal/mole. (**C**) Schematic representation of interactions between compounds and the TrkA.

**Figure 2 ijms-25-00206-f002:**
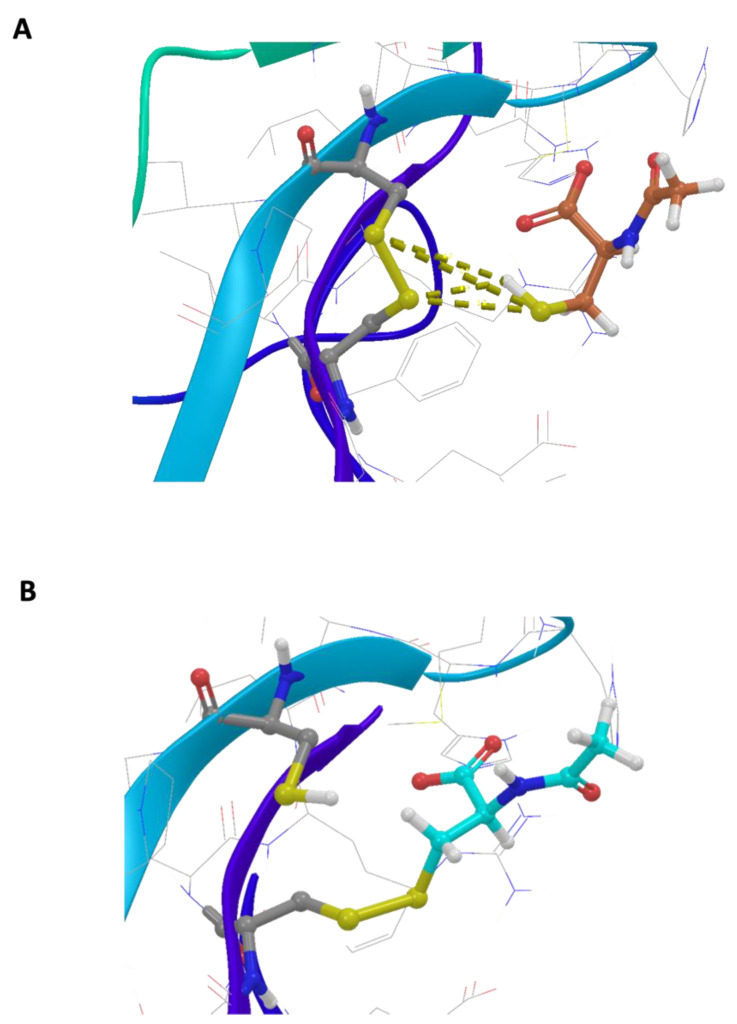
In silico analysis of the interaction between N-acetylcysteine and TrkA disulfide bond between cysteines 300–345. (**A**) Representative scheme of a H-bond involving both the Sp-Sp atoms. (**B**) Representative formation of the Sd-Sp bond between NAC and Cys345 and the protonation of sulfur atom of Cys300.

**Figure 3 ijms-25-00206-f003:**
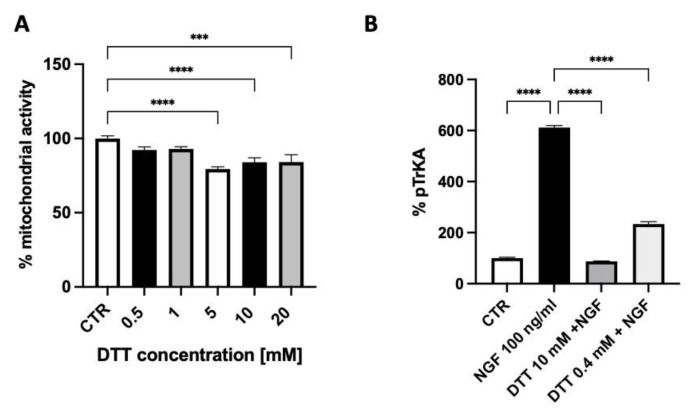
The effect of dithiothreitol (DTT) on cell mitochondrial activity (**A**) and on TrkA autophosphorylation elicited by NGF (**B**). Each value represents the mean ± S.E.M. of independent experiments with respect to the control (100%). **** *p* < 0.0001; *** ≤ 0.001; Dunnett’s multiple comparison post hoc test (**A**) and Tukey’s multiple comparison post hoc test (**B**); n = 3.

**Figure 4 ijms-25-00206-f004:**
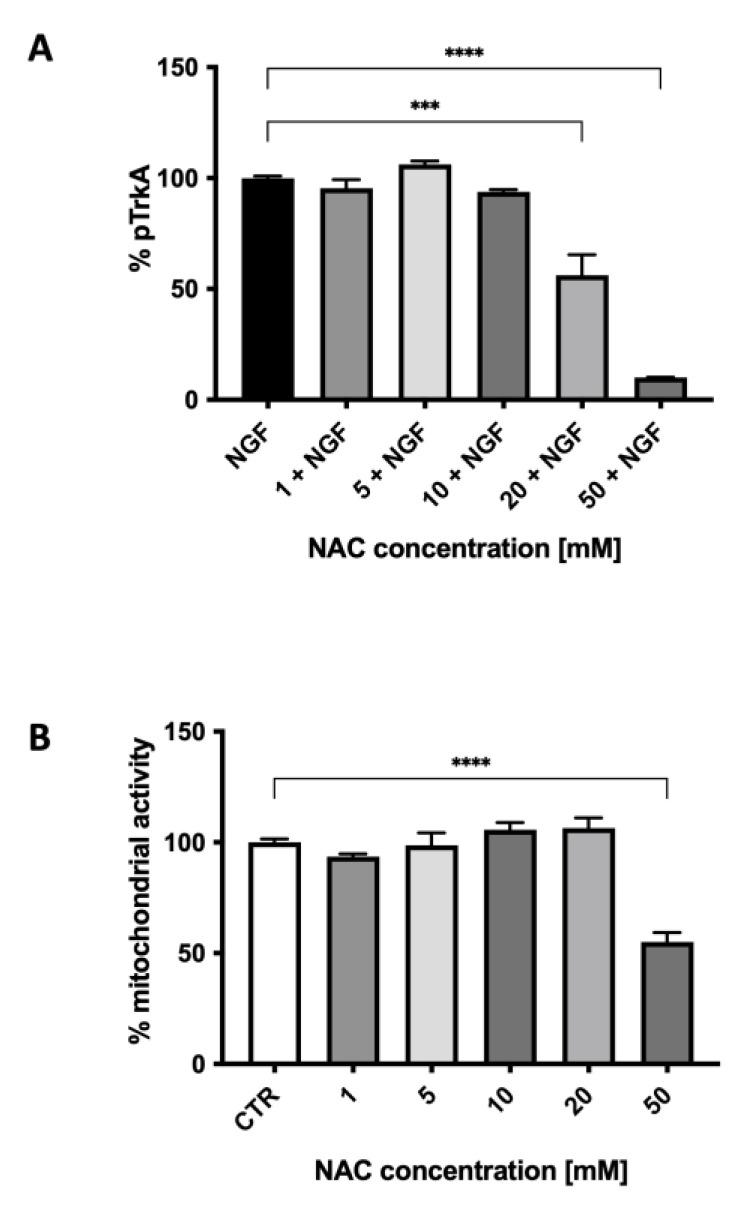
The effect of various concentrations of N-acetylcysteine on TrkA autophosphorylation elicited by NGF (**A**) and on cell mitochondrial activity (**B**). Each value represents the mean ± S.E.M. of independent experiments with respect to the control (100%). **** *p* < 0.0001; *** ≤ 0.001; Dunnett’s multiple comparison post hoc test; n = 3.

**Table 1 ijms-25-00206-t001:** Examples of the use of NAC to produce analgesic activity.

Condition	Notes and Reference
Diabetic neuropathy	In a mouse model [18]
Postoperative opioid consumption	Clinical trial in patients (Clinical Trial Registration: NCT04562597) [19]
Chronic pain	NAC in chronic pain management [20]
Endometriosis	Efficacy of N-Acetylcysteine on endometriosis [21]

## Data Availability

Data is contained within the article and Appendix A.

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
