# Peer review of "N-Acetylcysteine Antagonizes NGF Activation of TrkA through Disulfide Bridge Interaction, an Effect Which May Contribute to Its Analgesic Activity"

_ijms, 2023, doi:10.3390/ijms25010206_

Round 1

Reviewer 1 Report

Comments and Suggestions for Authors

The paper describes the possible mode of action of DTT polymers on TrkA. First and foremost, the title is somehow misleading "inhibition of activation" and would need to be changed. The study seems to has been thought thorough, yet I would like to point out some of, in my opinion, methodology inaccuracies regarding in silico study.
First and foremost, docking methodology – I couldn’t find whether the method has been validated and if so, what was the outcome, neither if this method was implemented and described earlier. The grid size is relatively large and false positive might be observed. Authors state that all the water molecules were removed. With the importance of water molecules in protein-ligand binding, a replaceability study of the water molecules ought to be performed.
From what I see on the Figure 1 the ligands do occur in the same binding space, yet the mode of docking is completely different. Also, main interacting AAs should be labelled, and the interactions represented with eg, dotted lines (even though it might be VdW interactions). Figure 2 shows the covalently bound ligand, and corresponding text says (lines 113-115): This also shows that the TrkA-NAC interaction is really able to break the Sp-Sp, thus producing a structural rearrangement internal to the bind-ing site and a consequent loss of the molecular recognition capability of TrkA- 115 NGF” - Was the rearrangement observed without a MD study? If so, a Figure demonstrating superimposition of starting state and rearranged protein structures should be attached. If not – such observation needs to be backed with at least ~250ns MD simulation followed by FEP simulations, so as to confirm those observations. Last but not least how was the global energy calculated. If only on the basis of the scoring function I’d highly suggest MM/GBSA calculations for the selected poses as well. Last but not least - in not patent restricted a Figure with structures of described ligands would be a great addition to the manuscript - when restricted, an appropriate statement should do. 

Some minor issues include:

Lines 50-55 sentence too long and barely understandable in this way

Chapter 2.1. results: HSd – please explain the abbreviation where appears

Lines 83 - The DTT species – unclear what species exactly ?

Lines 85-87 – “more and more” and “larger and larger” - repetitions

Lines 102 – TrkA-NAC literature reference?

Line 230 - pH 7.0 why so?

Comments on the Quality of English Language

The manuscript contains many verbatim translations from a foreign language into English, repetitions and multiple sentences making it difficult to read at times.

Author Response

REVIEWER 1

The paper describes the possible mode of action of DTT polymers on TrkA. First and foremost, the title is somehow misleading "inhibition of activation" and would need to be changed. The study seems to has been thought thorough, yet I would like to point out some of, in my opinion, methodology inaccuracies regarding in silico study.

As far as the title we changed it to eliminate the "inhibition of activation", we hope that now the title is clearer.

First and foremost, docking methodology – I couldn’t find whether the method has been validated and if so, what was the outcome, neither if this method was implemented and described earlier.

We thank the reviewer for the observation. Undoubtedly, it is good practice to assess the robustness of the docking protocol and the most accurate way to do so involves removing a co-crystallized ligand from the complex, re-docking such ligand and then calculating the root mean square deviation (RMSD) between the docked conformation and the native conformation (the smaller the RMSD, the better it is). Unfortunately, this is not feasible since there is no TrkA structure in the Protein Data Bank with a co-crystallized ligand other than NGF. However, the selected docking tool (Glide) and parameters are widely used and reliable, as demonstrated by studies reported in the literature.

  • Roy R, Sk MF, Tanwar O, Kar P. Computational studies indicated the effectiveness of human metabolites against SARS-Cov-2 main protease. Mol Divers. 2023 Aug;27(4):1587-1602. doi: 10.1007/s11030-022-10513-6. Epub 2022 Aug 18. PMID: 35978064; PMCID: PMC9385416.
  • Yao C, Shen Z, Shen L, Kadier K, Zhao J, Guo Y, Xu L, Cao J, Dong X, Yang B. Identification of Potential JNK3 Inhibitors: A Combined Approach Using Molecular Docking and Deep Learning-Based Virtual Screening. Pharmaceuticals (Basel). 2023 Oct 13;16(10):1459. doi: 10.3390/ph16101459. PMID: 37895928; PMCID: PMC10610115.
  • Krishnamoorthy HR, Karuppasamy R. A multitier virtual screening of antagonists targeting PD-1/PD-L1 interface for the management of triple-negative breast cancer. Med Oncol. 2023 Sep 30;40(11):312. doi: 10.1007/s12032-023-02183-7. PMID: 37777635.
  • Mali SN, Pandey A, Bhandare RR, Shaik AB. Identification of hydantoin based Decaprenylphosphoryl-β-D-Ribose Oxidase (DprE1) inhibitors as antimycobacterial agents using computational tools. Sci Rep. 2022 Sep 30;12(1):16368. doi: 10.1038/s41598-022-20325-1. PMID: 36180452; PMCID: PMC9525719.
  • Sandor M, Kiss R, Keserű GM. Virtual fragment docking by Glide: a validation study on 190 protein−fragment complexes. J Chem Inf Model. 2010. https://doi.org/10.1021/ci1000407.

The grid size is relatively large and false positive might be observed.

Thanks again for the suggestion, we know that a large grid size might affect the accuracy of the docking simulation. We have used a grid box with a size of 36Åx36Åx36Å (the max size allowed in Glide) to take into account the entire portion of TrkA that interact with NGF. In order to validate our results, we have redocked the ligands using a smaller box grid of 20ÅX20ÅX20Å which enclose the interaction surface between TrkA and NGF. The results are consistent with the previous ones; binding energies are nearly the same in both simulations, both in order of magnitude and their relative patterns. There are slight differences in the orientation of poses but all ligands are located in the same pocket in front of the disulfide bridge. Worth to note, the DTT compounds always interact with GLU331 through hydrogen bond (see table and figure).

Polymer DTT

Binding energy large box

(kcal/mol)

Binding energy small box

(kcal/mol)

ΔΔG (kcal/mol)

Tris-DTT

-5.4

-5.0

0.4

Bis-DTT

-4.5

-3.5

1.0

Mono-DTT

-3.2

-3,0

0.2

Authors state that all the water molecules were removed. With the importance of water molecules in protein-ligand binding, a replaceability study of the water molecules ought to be performed.

Thank you for the relevant comment. As mentioned in the previous comment and in the text, the simulations were carried out at the TrkA/NGF interface. No water molecules are present in this region inside the crystal structure. We have revised the text by omitting the sentence: “All water molecules were removed from the PDB file” (line 231).

From what I see on the Figure 1 the ligands do occur in the same binding space, yet the mode of docking is completely different. Also, main interacting AAs should be labelled, and the interactions represented with eg, dotted lines (even though it might be VdW interactions).

Thanks for the suggestion, in Figure 1 we have added panel C which illustrates the 2D binding pocket showing key amino acids and the interactions between the ligand and the protein.

Figure 2 shows the covalently bound ligand, and corresponding text says (lines 113-115): This also shows that the TrkA-NAC interaction is really able to break the Sp-Sp, thus producing a structural rearrangement internal to the bind-ing site and a consequent loss of the molecular recognition capability of TrkA- 115 NGF” - Was the rearrangement observed without a MD study? If so, a Figure demonstrating superimposition of starting state and rearranged protein structures should be attached. If not – such observation needs to be backed with at least ~250ns MD simulation followed by FEP simulations, so as to confirm those observations.

We thank the reviewer for constructive remarks. We do not directly observed/investigated the rearrangement after the disruption of disulfide bond. Considering the importance of disulfide bridges for maintaining protein folding (1,2,3), it is reasonable to assume that the elimination of this interaction could lead to a conformational rearrangement. In any case, even if the rearrangement is minimal or absent, the presence of a covalently bound molecule at the binding site would prevent the interaction between TrkA and NGF. The main purpose of this study is to identify compounds capable of preventing the binding of NGF to TrkA, without elucidating the mechanism of action. For this reason, we did not perform molecular dynamics simulations.

  • Zhou NE, Kay CM, Hodges RS. Disulfide bond contribution to protein stability: positional effects of substitution in the hydrophobic core of the two-stranded alpha-helical coiled-coil. Biochemistry. 1993 Mar 30;32(12):3178-87. doi: 10.1021/bi00063a033. PMID: 8457578.
  • Lakbub JC, Shipman JT, Desaire H. Recent mass spectrometry-based techniques and considerations for disulfide bond characterization in proteins. Anal Bioanal Chem. 2018 Apr;410(10):2467-2484. doi: 10.1007/s00216-017-0772-1. Epub 2017 Dec 18. PMID: 29256076; PMCID: PMC5857437.
  • Zhang L, Chou CP, Moo-Young M. Disulfide bond formation and its impact on the biological activity and stability of recombinant therapeutic proteins produced by Escherichia coli expression system. Biotechnol Adv. 2011 Nov-Dec;29(6):923-9. doi: 10.1016/j.biotechadv.2011.07.013. Epub 2011 Jul 29. PMID: 21824512.

Last but not least how was the global energy calculated. If only on the basis of the scoring function I’d highly suggest MM/GBSA calculations for the selected poses as well.

Thanks for the suggestion, we have calculated the binding energy using Prime MM-GBSA tool of Schrodinger suite.

Polymer DTT

Binding energy large box

(kcal/mol)

MM-GBSA

(kcal/mol)

Tris-DTT

-5.4

-89.8

Bis-DTT

-4.5

-71.5

Mono-DTT

-3.2

-38.1

Last but not least - in not patent restricted a Figure with structures of described ligands would be a great addition to the manuscript - when restricted, an appropriate statement should do.

Thanks for the suggestion, structures of described ligands are reported in figure 1 panel C

Minor Points

Lines 50-55 sentence too long and barely understandable in this way

The sentence was modified.

Chapter 2.1. results: HSd – please explain the abbreviation where appears

HSd stands for Thiol group docking species

Lines 83 - The DTT species – unclear what species exactly ?

We deleted species et this point, they are defined (monomers, dimers, trimers) in the following sentences,

Lines 85-87 – “more and more” and “larger and larger” – repetitions

We avoided the repetition in the revised version

Lines 102 – TrkA-NAC literature reference?

We are referring to the direct interaction described in this paper (fig. 2), we are not aware of other literature showing a direct interaction.

Line 230 - pH 7.0 why so?

We decided to use neutral pH as standard reference condition in our  calculations

Reviewer 2 Report

Comments and Suggestions for Authors

In this paper authors performed in silico and in vivo analysis of NAC antagonistic effect on TrkA receptor. These results are significant as an initial idea and statements that require further experimental research. The paper is well written and suitable for publication in this journal. However, I would like the authors to respond to the claim in the paper on Page 3, Line 100. Which value of binding energy indicate high binding affinity?

Author Response

REVIEWER 2

In this paper authors performed in silico and in vivo analysis of NAC antagonistic effect on TrkA receptor. These results are significant as an initial idea and statements that require further experimental research. The paper is well written and suitable for publication in this journal. However, I would like the authors to respond to the claim in the paper on Page 3, Line 100. Which value of binding energy indicate high binding affinity?

See the answers to referee 1 and the reported data in the tables of this letter and in the modified table of Fig. 1 in the manuscript. From a general point of view the more negative the DG the greater the affinity

Reviewer 3 Report

Comments and Suggestions for Authors

The paper describes additional mechanism of action of widely used mucolytic agent N-acetylcysteine (NAC). The authors first show that NAC could be used as a TrkA antagonist, an action that may contribute to the activity and use of NAC in various painful states (acute, chronic, nociplastic) sustained by nerve growth factor (NGF) hyperactivity. Despite very high concentrations of NAC necessary to demonstrate the effect (inhibition by 40% using 20 mM NAC), the results are interesting and could provoke further studies also in other groups.

The paper can be published after minor corrections of style and misprints (e.g. “biding energy”, line 111). References to the mucolytic effect of NAC should be added into Introduction section.

Comments on the Quality of English Language

In general English is very good, only a few sentences could be slightly corrected.

Author Response

REVIEWER 3

The paper describes additional mechanism of action of widely used mucolytic agent N-acetylcysteine (NAC). The authors first show that NAC could be used as a TrkA antagonist, an action that may contribute to the activity and use of NAC in various painful states (acute, chronic, nociplastic) sustained by nerve growth factor (NGF) hyperactivity. Despite very high concentrations of NAC necessary to demonstrate the effect (inhibition by 40% using 20 mM NAC), the results are interesting and could provoke further studies also in other groups.

The paper can be published after minor corrections of style and misprints (e.g. “biding energy”, line 111). References to the mucolytic effect of NAC should be added into Introduction section.

A sentence was added in the las part of introduction quoting the paper by Schwalfenberg et al. (2021) which reviews and quotes some of the original papers on the mucolytic properties of NAC and proposes a wide spectrum of potential  uses in other diseases.

Round 2

Reviewer 1 Report

Comments and Suggestions for Authors

The authors have addressed all the comments and have updated the manuscript accordingly